# A MAPK/miR-29 Axis Suppresses Melanoma by Targeting MAFG and MYBL2

**DOI:** 10.3390/cancers13061408

**Published:** 2021-03-19

**Authors:** Olga Vera, Ilah Bok, Neel Jasani, Koji Nakamura, Xiaonan Xu, Nicol Mecozzi, Ariana Angarita, Kaizhen Wang, Kenneth Y. Tsai, Florian A. Karreth

**Affiliations:** 1Department of Molecular Oncology, H. Lee Moffitt Cancer Center and Research Institute, Tampa, FL 33612, USA; olga.verapuente@moffitt.org (O.V.); ilah.bok@moffitt.org (I.B.); neel.jasani@moffitt.org (N.J.); kojinakamura@gyne.med.osaka-u.ac.jp (K.N.); xiaonan.xu@moffitt.org (X.X.); nicol.mecozzi2@moffitt.org (N.M.); aangari4@jhmi.edu (A.A.); Kaizhen.Wang@moffitt.org (K.W.); 2Cancer Biology PhD Program, University of South Florida, Tampa, FL 33612, USA; 3Department of Biology, University of Pisa, 56126 Pisa, Italy; 4Departments of Anatomic Pathology and Tumor Biology, H. Lee Moffitt Cancer Center and Research Institute, Tampa, FL 33612, USA; kenneth.tsai@moffitt.org; 5Donald A. Adam Melanoma and Skin Cancer Center of Excellence, Moffitt Cancer Center, Tampa, FL 33612, USA

**Keywords:** melanoma, melanocytes, miR-29, MAPK pathway, MAFG, MYBL2, BRAF, p53

## Abstract

**Simple Summary:**

The miR-29 family is subjected to complex regulation by tumor suppressors and oncogenes and has tumor suppressive potential in several cancers. We demonstrate that, in melanoma, oncogenic BRAF paradoxically induces miR-29 in concert with p53, thereby forming a barrier to melanoma progression. This barrier is overcome by reduced expression of miR-29, likely via diminished p53 activity. We further identify the transcription factors MAFG and MYBL2 as targets of miR-29 and show that their repression is detrimental for melanoma cells. Targeting MAFG- and MYBL2-regulated processes may therefore represent a promising therapeutic strategy to treat miR-29-low melanoma.

**Abstract:**

The miR-29 family of microRNAs is encoded by two clusters, miR-29b1~a and miR-29b2~c, and is regulated by several oncogenic and tumor suppressive stimuli. While in vitro evidence suggests a tumor suppressor role for miR-29 in melanoma, the mechanisms underlying its deregulation and contribution to melanomagenesis have remained elusive. Using various in vitro systems, we show that oncogenic MAPK signaling paradoxically stimulates transcription of pri-miR-29b1~a and pri-miR-29b2~c, the latter in a p53-dependent manner. Expression analyses in melanocytes, melanoma cells, nevi, and primary melanoma revealed that pri-miR-29b2~c levels decrease during melanoma progression. Inactivation of miR-29 in vivo with a miRNA sponge in a rapid melanoma mouse model resulted in accelerated tumor development and decreased overall survival, verifying tumor suppressive potential of miR-29 in melanoma. Through integrated RNA sequencing, target prediction, and functional assays, we identified the transcription factors MAFG and MYBL2 as bona fide miR-29 targets in melanoma. Our findings suggest that attenuation of miR-29b2~c expression promotes melanoma development, at least in part, by derepressing MAFG and MYBL2.

## 1. Introduction

While genetic and genomic alterations are established drivers of melanoma formation, aberrant control of gene expression is emerging as a major contributor to melanoma progression. microRNAs (miRNAs) bind to the 3′UTRs of target mRNAs to negatively regulate gene expression [1]. To date, various miRNAs have been reported to regulate the biology of melanoma [2,3] including miR-29a, for which melanoma suppressive potential has recently been proposed based on in vitro analyses [4]. The miR-29 family is encoded by two clusters, miR-29b1~a and miR-29b2~c, that produce three mature miRNAs, miR-29a, miR-29b, and miR-29c [5,6]. The mature miR-29 family members are highly conserved across species and share identical seed sequences [6]. miR-29 is considered a tumor suppressor miRNA given its ability to repress genes involved in proliferation and cell survival, such as AKT3 [7,8], DNMT3A/B [9], MCL1 [10], and CDK6 [11], and its frequent downregulation in cancer [5], including leukemia [12], ovarian [13], or breast. Interestingly, miR-29 has emerged as a major regulatory hub that integrates signaling from potent oncogenes and tumor suppressors. Indeed, miR-29 expression is repressed by the oncogenes c-Myc, Hedgehog, and NF-κB [14,15]. NRF2 was reported to stimulate or suppress miR-29 expression depending on the cell type [16,17], and p53 promotes miR-29 expression when stimulated by aging or chronic DNA damage [8]. However, the regulation of miR-29 and its function in melanoma biology are poorly understood.

In this study, we examined if, similar to other oncogenes, mutant BRAF opposes the expression of miR-29. Surprisingly, we found that acute MAPK pathway activation increases expression of both miR-29b1~a and miR-29b2~c, the latter in a p53-dependent manner. However, our results suggest that miR-29b2~c expression is attenuated during melanoma development, possibly as a consequence of diminished p53 activity. Inactivation of miR-29 in a melanoma mouse model augmented tumor development. Finally, we identified MAFG and MYBL2 as miR-29 targets whose de-repression may be critical for melanoma development.

## 2. Results

### 2.1. Oncogenic BRAF Promotes miR-29 Expression in MEFs

Given the regulation of miR-29 by several cancer-associated pathways, we first examined if BRAF^V600E^ modulates miR-29 levels in mouse embryonic fibroblasts (MEFs) carrying a Cre-inducible endogenous Braf^V600E^ allele (LSL-Braf^V600E^) [18]. In contrast to other oncogenic pathways, adenoviral-Cre mediated activation of endogenous Braf^V600E^ increased mature miR-29a, -29b, and -29c levels (Appendix A). To determine if the regulation of miR-29 occurs at the transcriptional level, we assessed the expression of the primary pri-miR-29b1~a and pri-miR-29b2~c transcripts by TaqMan qRT-PCR assay. Endogenous activation of oncogenic Braf^V600E^ increased both pri-miR-29b1~a and pri-miR-29b2~c (Appendix A), indicating that BRAF^V600E^ promotes the transcription of both miR-29 clusters.

p53 has been shown to promote expression of mature miR-29a, -29b, and -29c [8], and we thus investigated whether oncogenic BRAF regulates miR-29 by provoking p53 activity. To validate p53-dependent regulation of miR-29, we treated wildtype MEFs with the DNA-damaging agents Doxorubicin and Mitomycin C. While DNA damage induced p53 as expected, we surprisingly observed that Doxorubicin and Mitomycin C induced the transcription of only pri-miR-29b2~c (Appendix A). The previously reported increased expression of mature miR-29a in response to p53 activity is likely explained by the inability of the mature miRNA qRT-PCR assay to distinguish between miR-29a and miR-29c, which only differ in one nucleotide, as has been suggested previously [16]. Indeed, miR-29a and miR-29c Taqman qRT-PCR assays were unable to distinguish between miR-29a and miR-29c mimics transfected into A375 melanoma cells (Appendix A). Thus, p53 regulates expression of only miR-29b2~c, while oncogenic BRAF induces transcription of both miR-29b1~a and miR-29b2~c.

We next examined the involvement of the MAPK pathway and the dependence of miR-29 regulation on p53. We knocked down p53 in LSL-Braf^V600E^ MEFs (Appendix A) and treated control and p53 silenced cells with the MEK inhibitor AZD6244 for 24 h. AZD6244 treatment decreased basal and Braf^V600E^-induced expression of pErk and the MAPK pathway transcriptional target c-Jun in the presence or absence of p53 (Appendix A). pri-miR-29b1~a expression was similarly reduced by MEK inhibition in control and p53 silenced MEFs (Appendix A), indicating that oncogenic BRAF regulates miR-29b1~a independently of p53 via the MAPK pathway. AZD6244 treatment blunted Braf^V600E^-induced pri-miR-29b2~c expression, possibly due to reduced p53 expression (Appendix A). Notably, neither Braf^V600E^ expression nor MEK inhibition affected pri-miR-29b2~c levels in the absence of p53 (Appendix A), indicating that p53 is essential for the expression of this miR-29 cluster. These data suggest differential regulation of the two miR-29 clusters by Braf^V600E^: while miR-29b1~a is controlled by the MAPK pathway, miR-29b2~c expression depends on p53.

### 2.2. The MAPK Pathway Regulates miR-29 Expression in Melanocytes and Melanoma Cells

We next examined whether the regulation of miR-29 observed in MEFs also occurs in melanocytes. Induction of Braf^V600E^ by adenoviral Cre in two primary mouse melanocyte cultures isolated from LSL-Braf^V600E^ mice increased the expression of both miR-29 clusters (Figure 1A), suggesting that the regulation of miR-29 by oncogenic BRAF is conserved in fibroblasts and melanocytes. To analyze miR-29 regulation by the MAPK pathway in melanocytes, we starved Hermes1 and Hermes3A cells of TPA, a phorbol ester that stimulates the MAPK pathway by activating PKC [19] and that is required for melanocyte proliferation in vitro. Re-stimulation with TPA increased pERK levels and promoted transcription of pri-miR-29b1~a and pri-miR-29b2~c (Figure 1B). Furthermore, MEK inhibition in Hermes1 and Hermes3A cells cultured in the presence of TPA diminished the expression of pri-miR-29b1~a and pri-miR-29b2~c (Figure 1C). To further analyze miR-29 regulation during melanocyte transformation, we examined the consequences of chronic BRAF^V600E^ expression in human melanocytes. We delivered lentiviral HA-tagged BRAF^V600E^ to Hermes1 and Hermes3A cells, which resulted in the emergence of four independent clones, one from Hermes1 (H1B) and three from Hermes3A (H3B2, H3B4, and H3B8). These cell lines express ectopic BRAF^V600E^ and exhibit increased MAPK signaling as shown by elevated pERK and c-JUN (Appendix A), which enables these cells to proliferate in the absence of TPA (Appendix A). Expression of both pri-miR-29b1~a and pri-miR-29b2~c was sensitive to AZD6244 treatment in H1B and H3B8 cells (Figure 1D). Finally, AZD6244 also decreased the levels of pri-miR-29b1~a in a panel of human melanoma cell lines, while pri-miR-29b2~c was only moderately reduced in three out of eight cell lines (Figure 1E,F). These findings indicate that, while the MAPK pathway promotes expression of both miR-29 clusters in melanocytes, the miR-29b2~c cluster may become less reliant on MAPK signaling during melanocyte transformation and melanoma formation.

We hypothesized that BRAF^V600E^-induced expression of tumor suppressive miR-29 may constrain melanoma development, prompting downregulation of miR-29 during melanomagenesis. To analyze miR-29 levels during melanomagenesis, we measured pri-miR-29b1~a and pri-miR-29b2~c expression in four wildtype BRAF human melanocyte cell lines (Hermes1, Hermes2, Hermes3A, and Hermes4B), four melanocyte cell lines stably expressing BRAF^V600E^ (H1B, H3B2, H3B4, H3B8), and eleven human melanoma cell lines (A375, SkMel28, WM35, WM266.4, WM115, WM164, 451Lu, WM793, 1205Lu, SbCl2 and 501Mel) by qRT-PCR. pri-miR-29b1~a expression varied considerably between cell lines with a trend toward reduced expression in melanoma cells compared to BRAF-mutant melanocytes (Figure 1G). By contrast, pri-miR-29b2~c levels significantly decreased in BRAF-mutant melanocytes and melanoma cells compared to wildtype melanocytes (Figure 1G). In addition, we interrogated miR-29 expression in a publicly available RNAseq dataset [20] containing 23 nevi and 57 primary melanomas. We found a trend towards increased pri-miR-29b1~a expression in primary melanomas compared to nevi, while pri-miR-29b2~c levels were significantly reduced (Figure 1H). Thus, diminished expression of pri-miR-29b2~c is associated with the progression of transformed melanocytes to frank melanoma.

Given the critical dependence of miR-29b2~c transcription on p53 in MEFs, we determined whether the decrease of pri-miR-29b2~c during melanoma formation is related to p53 activity. Similar to MEFs, treatment of Hermes1 and Hermes3A melanocytes with Doxorubicin robustly increased protein expression of p53 and its target p21 (Appendix A). Doxorubicin also enhanced pri-miR-29b2~c expression (Appendix A), indicating that p53 regulates pri-miR-29b2~c also in melanocytes. Conversely, the induction of p53 and/or p21 was diminished or absent in 8 out of 11 human melanoma cell lines treated with Doxorubicin (Appendix A). Moreover, all four BRAF-mutant melanocyte cell lines lost the expression of p53 (Appendix A). Thus, reduced expression of pri-miR-29b2~c in BRAF-mutant melanocytes and in melanoma cell lines correlated with impaired p53 activity.

### 2.3. miR-29 Inactivation Promotes Melanoma Formation

To examine if reduced expression of miR-29 promotes melanoma formation, we first tested a miRNA sponge approach to inactivate miR-29 in vitro. To this end, A375 melanoma cells were either transfected with a hairpin inhibitor of miR-29a or transduced with a lentiviral bulged miR-29 sponge construct. The miR-29 sponge enhanced the colony formation capacity of A375 cells similar to the miR-29 hairpin inhibitor (Appendix A). In addition, the miR-29 sponge increased the activity of a miR-29 Luciferase reporter (Appendix A). Given the extensive overlap of predicted miR-29 targets in human and mouse, we used this miR-29 sponge construct in combination with our melanoma mouse modeling platform [21] to assess the effect of miR-29 inactivation in melanoma development. We targeted GEMM-derived Braf^V600E^; Pten^FL/WT^ embryonic stem cells with a Doxycycline (Dox)-inducible, GFP-linked miR-29 sponge allele or GFP as a control and produced miR-29 sponge and GFP experimental chimeras (Appendix A). Notably, chimeras expressing the miR-29 sponge developed melanoma with shorter latency (Figure 2A) and exhibited reduced overall survival (Figure 2B), indicating that inactivation of miR-29 accelerates melanoma development. Moreover, while all control mice developed only one melanoma, 37.5% of miR-29 sponge mice developed more than one tumor (Figure 2C). All tumors expressed the melanoma marker S100, confirming their melanocytic origin (Figure 2D). Histologically, GFP control tumors are characterized by small spindled cells often with loose edematous stroma while miR-29 sponge tumors are more punctuated by hypercellular areas with readily-appreciated mitoses. However, immunostaining of Ki67 revealed no difference in melanoma cell proliferation (Figure 2D,E), suggesting that miR-29 inactivation has more pronounced effects on tumor initiation than on progression. Accordingly, we did not observe any gross metastases in miR-29 sponge or GFP control mice.

We derived a melanoma cell line from a miR-29 sponge chimera to validate the functionality of the miR-29 sponge in mice and to examine the effects of restoring miR-29 activity. Turning off miR-29 sponge expression by withdrawing Dox from the culture resulted in repression of a miR-29 Luciferase reporter (Figure 2F), confirming that the sponge inactivated endogenous miR-29. Moreover, Dox withdrawal reduced proliferation (Figure 2G) and colony formation (Figure 2H) of the miR-29 sponge melanoma cells, indicating that continued miR-29 inactivation supports the transformed state.

### 2.4. Melanoma Cells Are Addicted to High Levels of miR-29 Target Genes

miR-29 may elicit its tumor suppressive potential by repressing targets such as AKT3, DNMT3A/B, or MCL1 [7,8,9,10]; however, miR-29 hairpin inhibitors failed to increase the expression of these validated targets in A375 melanoma cells (Figure 3A). Thus, alternative miR-29 targets must be responsible for the observed phenotypes in human cells and our miR-29 sponge melanoma mouse model. To identify melanoma-relevant miR-29 targets, we transfected A375 cells with miR-29a mimics and performed RNA sequencing. 6358 genes were differentially expressed in response to miR-29a mimics, and of those 1309 were significantly downregulated (Log2FC < −0.5 and FDR < 0.05). We further prioritized potential miR-29 targets based on three criteria: (i) increased expression in primary melanoma compared to nevi in the GSE112509 dataset with a Log2FC ≥ 0.3 and a FDR ≤ 0.05, (ii) the presence of high-confidence conserved miR-29 binding sites predicted by at least eight different algorithms, and (iii) a negative correlation in expression with pri-miR-29b2~c in the GSE112509 dataset with r ≤ −0.3 (Figure 3B). This analysis yielded nine candidate target genes: KCTD5, MYBL2, SLC31A1, MAFG, RCC2, TUBB2A, SH3BP5L, SMS, and NCKAP5L (Figure 3C,D). Analyzing the TCGA-SKCM dataset revealed that high expression of the nine identified candidate miR-29 targets is associated with poorer survival of melanoma patients (Figure 3E), suggesting oncogenic roles for these miR-29 targets in melanoma.

### 2.5. MAFG and MYBL2 Are Putative miR-29 Targets with Roles in Melanoma

To reveal which candidate miR-29 targets affect melanoma biology, we first tested whether miR-29 regulated their expression. Restoring miR-29 activity by withdrawing Dox from the murine miR-29 sponge melanoma cell line decreased the expression of 8 out of the 9 identified targets (Figure 4A). Moreover, miR-29 miRNA mimics decreased expression of all nine targets, while inhibition of endogenous miR-29 with hairpin inhibitors increased expression of 7 out of 9 targets in the murine miR-29 sponge melanoma cells (Figure 4B) and human A375 cells (Figure 4C). Thus, expression of the identified candidate genes is affected by modulation of miR-29 levels. miR-29 most likely suppresses melanoma development by repressing target mRNAs. We therefore tested if the identified miR-29 candidate targets are critical for melanoma cell proliferation. To this end, we individually silenced the nine candidate miR-29 targets using siRNA pools (Figure 4D) and performed proliferation and colony formation experiments. Knockdown of *MAFG* and, to a lesser extent, *MYBL2* significantly decreased proliferation (Figure 4E) and focus formation (Figure 4F,G) of A375 melanoma cells, suggesting that miR-29 may suppress melanoma by targeting these two genes.

We next sought to validate MAFG and MYBL2 as targets of miR-29 in melanoma. MYBL2 has previously been described as a miR-29 target [22,23], and to corroborate this finding we created a MYBL2 3′UTR Luciferase reporter and mutated the seed sequence of the miR-29 binding site (Appendix A). Co-transfection of miR-29 inhibitor with the wildtype MYBL2 3′UTR reporter into A375 and WM164 cells increased Luciferase activity (Appendix A), whereas miR-29 mimics reduced the activity of the wildtype MYBL2 3′UTR reporter. This effect was partially rescued by the miR-29 binding site mutation (Appendix A), indicating that MYBL2 is a direct target of miR-29 also in melanoma.

### 2.6. MAFG Is a Bona Fide Target of miR-29 in Melanocytes and Melanoma

MAFG has not been described as a miR-29 target. Interestingly, however, MAFG is an epigenetic regulator and transcriptional repressor in melanoma and hyperactive MAPK may increase MAFG stability by ERK-mediated phosphorylation [24]. Given these observations and the significant effect of MAFG silencing on melanoma cell growth, we selected MAFG for further analysis. To validate MAFG as a target of miR-29, we transfected miR-29a, miR-29b, or miR-29c mimics into BRAF^V600E^-expressing melanocytes (H1B) and melanoma cells (WM164). We observed a general reduction of MAFG mRNA and protein expression (Figure 5A,B). By contrast, hairpin inhibitors of miR-29a, miR-29b, or miR-29c increased MAFG mRNA and protein levels in these cell lines (Figure 5A,B). Next, similar to MYBL2, we generated a MAFG 3′UTR Luciferase reporter (Appendix A). We observed that co-transfection of miR-29 inhibitor with the MAFG 3′UTR reporter into H1B, H3B8, A375, and WM164 cells increased Luciferase activity (Figure 5C and Appendix A). Conversely, miR-29 mimics decreased the activity of the MAFG 3′UTR reporter (Figure 5D and Appendix A). We then mutated the seed sequence of the miR-29 binding site with the highest prediction score in the MAFG 3′UTR reporter and found that the effect of the miR-29 mimics was rescued by mutating the miR-29 binding site (Figure 5D and Appendix A). In addition, we observed slightly increased MAFG protein levels in bulk melanomas from miR-29 sponge chimeras compared to GFP control mice (Figure 5E). These findings indicate that MAFG is a bona fide target of miR-29.

We next assessed if the expression of MAFG is altered during melanoma development. First, we analyzed MAFG expression in melanocytes and melanoma cell lines and found that MAFG mRNA levels are increased in melanoma cell lines (Figure 5F). MAFG protein expression was similarly elevated in melanoma cell lines compared to melanocytes (Figure 5G,H). Notably, acute activation of endogenous Braf^V600E^ in primary melanocytes diminished *MAFG* mRNA expression (Figure 5I), which correlated with increased miR-29 expression (Figure 1A). Conversely, *MAFG* mRNA levels were increased in BRAF^V600E^-expressing Hermes cells in which p53 is lost and pri-miR-29b2~c is reduced (Figure 5J). This increase in *MAFG* mRNA led to a robust increase in MAFG protein expression that is more pronounced than stabilizing MAFG protein through TPA-induced MAPK signaling (Figure 5K). These results indicate that deregulation of MAFG during melanoma progression occurs through different mechanisms that promote increased expression of MAFG at mRNA and protein levels.

## 3. Discussion

Deregulation of miRNAs frequently occurs in cancer and is thought to play critical roles in all aspects of tumorigenesis. Here, we investigated the deregulation of miR-29 in melanoma formation. Using MEFs and human melanocytes, we uncovered paradoxical upregulation of miR-29b1~a by oncogenic BRAF, while MAPK signaling and p53 act in concert to promote miR-29b2~c expression. Diminished expression of the p53-dependent miR-29b2~c cluster is associated with the progression to frank melanoma, and inactivation of miR-29 promotes melanoma development in mice. De-repression of MAFG and MYBL2, which we identified as bona fide targets of miR-29, may contribute to melanoma development.

Previous studies have shown that p53 regulates the expression of both miR-29b1~a and miR-29b2~c [8,25]. However, our results indicate that transcription of miR-29b1~a is independent of p53, both in MEFs and in melanocytes. Instead, miR-29b1~a is regulated directly via MAPK signaling. This discrepancy is likely due to the fact that mature miR-29 species were analyzed by qRT-PCR in the previous studies, a method that failed to distinguish mature miR-29 family members, as has been suggested previously [16]. We also observed regulation of miR-29b2~c by MAPK signaling; however, this only occurs in the presence of p53. It remains to be investigated how MAPK signaling and p53 activation coordinately enhance miR-29b2~c expression upon acquisition of an oncogenic BRAF mutation. p53 may be induced by oncogene-activated MAPK pathway hyperactivation as ERK has been reported to phosphorylate p53 at serine 15 [26]. MAPK hyperactivation downstream of oncogenic BRAF is a critical driver of melanoma development [27], and p53 activation in response to mutant BRAF has been observed in melanocytes [28,29]. Given that oncogenic BRAF only very moderately activates p53 [30,31], BRAF may work in concert with rather than through p53.

Several reports describe tumor suppressive functions for miR-29 in cultured cells, including in melanoma cell lines [4,9,25], which we corroborated in our study. Given the tumor suppressive functions of miR-29 and its regulation by MAPK signaling, we hypothesized that MAPK hyperactivation could provoke a miR-29-dependent tumor suppressor response that prevents melanoma formation. The MAPK pathway is almost universally hyperactivated in melanoma, owing to the frequent activating mutations in BRAF and NRAS [27,32,33,34]. Notably, growth arrested nevi are common in humans and >80% of nevi harbor BRAF^V600E^ mutations [35,36], indicating the existence of potent tumor suppressive mechanisms [37]. To overcome this barrier, BRAF/NRAS mutant melanocytes might reverse the increase in miR-29 levels. We observed that, compared to melanocytes, miR-29b2~c expression is decreased in BRAF^V600E^-mutant, p53-deficient melanocytes and in melanoma cell lines. Similarly, miR-29b2~c is reduced upon the progression from nevi to primary melanomas. It is tempting to speculate that, while miR-29b1~a remains elevated due to continuous MAPK hyperactivation, impaired p53 activity leads to decreased miR-29b2~c expression, thereby promoting progression from nevi to frank melanoma. p53 may play a role in the growth arrest of nevi [28,38,39], and p53 inactivation in genetically engineered mice promotes melanoma development in the context of BRAF^V600E^ [40]. p53 is inactivated in melanoma through mutations and copy number losses [41,42,43,44], deletions of CDKN2A [41,42,43,44], or amplifications of MDM2 [45,46], all of which could lead to a reduction in miR-29b2~c expression. Thus, the inactivation of p53 may promote melanoma development in part by reducing the levels of miR-29b2~c.

Using a high-throughput mouse modeling approach, we found that sponge-mediated inactivation of miR-29 specifically in Braf^V600E^; Pten^Δ^**^/^**^WT^ melanocytes accelerated the development of melanoma. This is the first model used to study miR-29 inactivation in tumorigenesis, and also affirmed that synthetic miRNA sponges are powerful tools to examine miRNA function in vivo. One advantage over traditional modeling approaches, such as the previously published conditional knock-out allele of miR-29b1~a [47], is that a miRNA sponge has the potential to inactivate all members of a miRNA family. However, it is usually not clear how well a sponge interacts with each family member, especially in cases like the miR-29 family where one member, miR-29b, also localizes to the nucleus [48]. Thus, future studies using alternative approaches such as CRISPR/Cas9-mediated specific deletion of individual clusters or miRNAs will further elucidate the role of each miR-29 family member in melanoma. Moreover, while our findings ascertain a tumor suppressive function of miR-29 in melanoma, future studies will reveal the stages at which miR-29 suppresses melanoma development.

Since we did not observe changes in the expression of the validated miR-29 targets AKT3, MCL1, and DNMT3B, we identified new targets whose repression may contribute to restricting melanoma development. Of the nine genes identified by our approach, RCC2, MYBL2, and SLC31A1 had previously been identified as miR-29 targets [22,49,50]. While all nine genes were validated as miR-29 targets and the miR-29 sponge modulated the expression of eight of these targets, silencing of only MAFG and MYBL2 diminished the proliferation of melanoma cells.

MYBL2 has been described as a target of miR-29 during senescence of HeLa cells [22], and we confirmed that miR-29 represses MYBL2 also in melanoma cells. Moreover, we validated MAFG as bona fide target of miR-29 in melanoma. Interestingly, the MAFG protein is stabilized by ERK-mediated phosphorylation [24], suggesting that MAPK signaling converges on MAFG via ERK and miR-29. In addition to being repressed by miR-29, TCGA data indicate copy number gains of MAFG in melanoma. Thus, MAFG is deregulated in melanoma through multiple mechanisms and miR-29-mediated repression of MAFG may impair the transition from nevi to frank melanoma. Oncogenic roles for MAFG have so far been described in lung, ovarian, colorectal, and liver cancer [24,51,52,53]. In melanoma, oncogenic BRAF^V600E^ has been shown to stabilize MAFG, resulting in the recruitment of an epigenetic repressor complex to promoters and transcriptional silencing [24]. However, whether this epigenetic gene expression regulation and/or additional transcription factor complexes involving MAFG play critical roles in melanoma formation is unknown. MYBL2 has oncogenic properties in breast, lung, colorectal [54,55,56,57] and other cancers (reviewed in [58]), and our findings suggest a role for MYBL2 also in melanoma. Future studies will address if overexpression of MAFG or MYBL2 contributes to melanocyte transformation and melanoma development.

## 4. Materials and Methods

### 4.1. Cell Culture and Treatments

Hermes1, Hermes2, Hermes3A, and Hermes4B were cultured according to the protocol described by the Wellcome Trust Functional Genomics Cell Bank (https://www.sgul.ac.uk/about/our-institutes/molecular-and-clinical-sciences/research-centres/cell-biology-research-centre/genomics-cell-bank). Hermes1 and Hermes3A expressing BRAF^V600E^ were grown in the absence of TPA. Human cancer cell lines were cultured in RPMI containing 5% FBS at 37 °C in a humidified atmosphere containing 5% CO_2_. MEFs and HEK293T Lenti-X were cultured in DMEM containing 10% FBS at 37 °C in a humidified atmosphere containing 5% CO_2_. Primary mouse melanocytes were isolated as previously described [59]. To recombine floxed alleles, Mouse embryonic Fibroblasts (MEFs) and Primary Mouse Melanocytes (PMM) were infected with Ad5CMVCre or Ad5CMVempty adenovirus obtained from the University of Iowa Viral Vector Core (https://medicine.uiowa.edu/vectorcore/). All cell lines were routinely tested for mycoplasma using MycoAlert Plus (Lonza, Basel, Switzerland, Cat # LT07-710), and human melanoma cell lines were STR authenticated by Moffitt’s Molecular Genomics Core. Doxorubicin (Fisher Scientific, Waltham, MA, USA, Cat # BP25131) was used at 10 µM for 24 h and AZD6244 (Selleckchem, Houston, TX, USA, Cat # S1008) was used at 0.5 µM for 8 or 24 h. Detailed procedures are available in Appendix A. The mutation status of the melanoma cell lines used in the study is detailed in Appendix A.

### 4.2. RNA Isolation and Quantitative RT-PCR

Total RNA and mature miRNAs were isolated, reverse transcribed and quantitatively analyzed by qRT-PCR as previously described [21,51]. Samples were analyzed in triplicate using the StepOne Plus PCR system (Applied Biosystems, Foster City, CA, USA). snoU6 was used as endogenous control for mature miRNAs while GAPDH or β-Actin were used for mRNAs and pri-miRNAs. Taqman probes and primers for SYBR Green qPCR are listed in Appendix A.

### 4.3. RNA-Sequencing

RNA extraction, assessment of quality, library preparation, normalization, and analysis of RNAseq performed by Novogene are described in Appendix A. Accession number: PRJNA624657 (https://www.ncbi.nlm.nih.gov/sra; 15 March 2021)

### 4.4. Plasmids, Cell Transfection, and Lentiviral Transduction

Plasmid generation is described in the Appendix A. Cells were transfected with 25 to 150 nM of Dharmacon miRIDIAN microRNA miR-29a, miR-29b or miR-29c mimic (Fisher Scientific, Waltham, MA, USA, C-310521-07-0002, C-310381-05-0002 or C-310522-05-0002), hairpin inhibitor (Fisher Scientific, Waltham, MA, USA, IH-310521-08-0002, IH-310381-07-0002 or IH-310522-08-0002), or negative controls (Fisher Scientific, Waltham, MA, USA, CN-002000-01-05; IN-001005-01-05) using JetPrime (VWR, Radnor, PA, USA, Cat # 89129-924). For Luciferase assays, cells were co-transfected with either MAFG or MYBL2 psiCHECK2-3′UTR_wildtype or psiCHECK2-3′UTR_miR-29-mutant and miR-29 mimics or inhibitors. Luminescence was assayed after 24 h using the Dual Luciferase Assay System (Promega, Madison, WI, USA, Cat # E1960). Standard procedures were followed for retrovirus and lentivirus production and infection. For siRNA transfections, cells were transfected with 25 nM of ON-TARGETplus siRNA pools or Non-Targeting control using JetPrime. siRNA pool catalog numbers are available in Appendix A.

### 4.5. Proliferation and Colony Formation Assays

For proliferation assays, cells were plated in 96-well plates at a density of 1000–2500 cells/well. Cells were fixed and stained with 0.1% crystal violet (VWR, Radnor, PA, USA, Cat # 97061-850) in 20% methanol. Crystal violet was extracted with 10% acetic acid and absorbance measured at 600 nm. For colony formation assays, cells were plated in 6-well plates at a density of 1000–2000 cells/well and cultured for 2–3 weeks. Cells were fixed and stained with 0.1% crystal violet as above and colonies were quantified using ImageJ v1.53c.

### 4.6. Immunoblotting

Protein isolation was performed as previously described [21]. In addition, 20 μg of total protein were subjected to SDS-PAGE and Western blot as described previously [21]. Primary antibodies used and detailed procedures are available in Appendix A. Appendix A shows the uncropped Western Blots used for this manuscript.

### 4.7. ES Cell Targeting, Mouse Generation, and ESC-GEMM Experiments

ES cell targeting and generation of chimeras was performed a described previously [21]. Melanoma development was induced in 3–4-week-old chimeras having similar ESC contribution using 25 mg/mL 4-OH Tamoxifen Mice were fed 200 mg/kg Doxycycline (Envigo, Indianapolis, IN, USA, Cat # TD180625) ad libitum. All animal experiments were conducted in accordance with an IACUC protocol approved by the University of South Florida.

### 4.8. Statistical Analysis

Statistical analysis was performed using GraphPad Prism 8.3 (https://www.graphpad.com/scientific-software/prism/). Survival data were compared by applying the Gehan-Breslow-Wilcoxon test, and all other data were analyzed with the unpaired two-tailed *t*-test or ordinary one-way ANOVA. A *p*-value below 0.05 was considered statistically significant. Experiments were performed in triplicates or quadruplicates and each experiment was repeated at least once. Unless otherwise indicated, one representative experiment is shown. Data represent the mean ± SEM.

## 5. Conclusions

Our work uncovered that miR-29 constrains MAPK pathway-driven melanoma formation, at least in part, by repressing MAFG and MYBL2. Although there are no current clinical trials that explore the effect of miRNA mimics in melanoma, the use and safety of miR-29 mimics (remlarsen, MRG-201) have been previously tested as a treatment to prevent skin fibrosis (Clinical Trials.gov Identifier: NCT02603224, [60]). Therefore, our in vivo validation that miR-29 is a tumor suppressive miRNA in melanoma might contribute to the development of new therapeutic approaches, not only targeting MAFG or MYBL2, but also miR-29 itself.

## Figures and Tables

**Figure 1 cancers-13-01408-f001:**
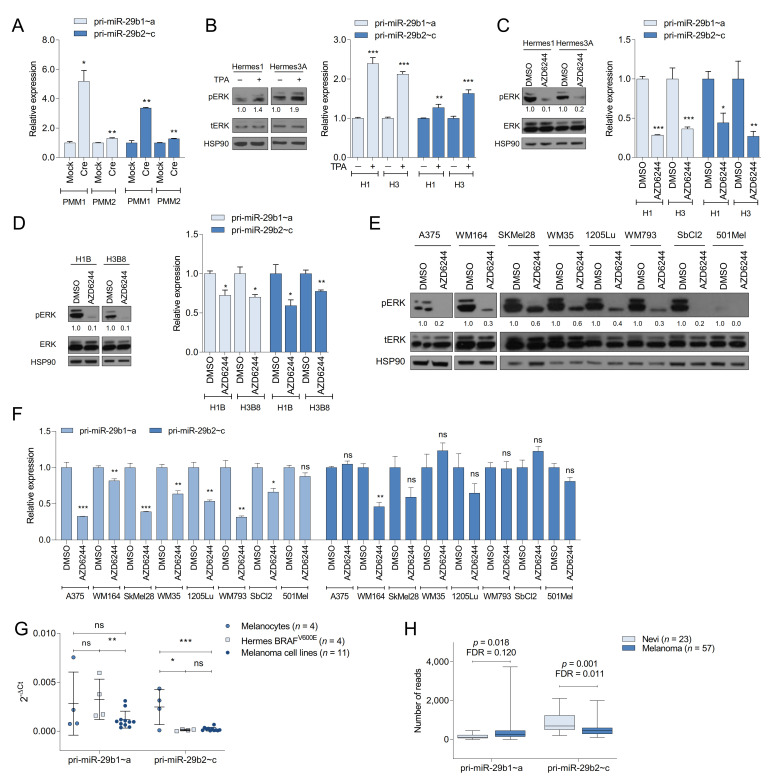
MAPK signaling and p53 regulate miR-29 in human melanocytes and melanoma. (**A**) qRT-PCRs showing the expression of pri-miR-29b1~a and pri-miR-29b2~c in LSL-Braf^V600E^ primary mouse melanocytes (PMM) following Adeno-Cre/Mock infection; (**B**) effect of TPA on pri-miR-29b1~a and pri-miR-29b2~c expression in human melanocytes; (**C**) effect of AZD6244 on pri-miR-29b1~a and pri-miR-29b2~c expression in human melanocytes; (**D**) effect of AZD6244 on pri-miR-29b1~a and pri-miR-29b2~c expression in H1B and H3B8 cells; (**E**,**F**) effect of AZD6244 on pri-miR-29b1~a and pri-miR-29b2~c expression in human melanoma cells; (**G**) expression of pri-miR-29b1~a and pri-miR-29b2~c in human melanocytes (*n* = 4), BRAF^V600E^ melanocytes (*n* = 4) and melanoma cell lines (*n* = 11); (**H**) expression of pri-miR-29b1~a and pri-miR-29b2~c in nevi (*n* = 23) and melanoma (*n* = 57) in the GSE112509 dataset. pri-miR-29b1~a and pri-miR-29b2~c qRT-PCRs are shown in the right panels while Western blots are shown in the left panels of (**B**–**D**). The mean ± SEM of one representative out of two independent experiments performed in triplicates is shown. RNA expression is normalized to β-Actin. All Western blots show the intensity ratio of the protein of interest normalized to HSP90. ns, not significant; FDR, false discovery rate; * *p* < 0.05; ** *p* < 0.01; *** *p* < 0.001.

**Figure 2 cancers-13-01408-f002:**
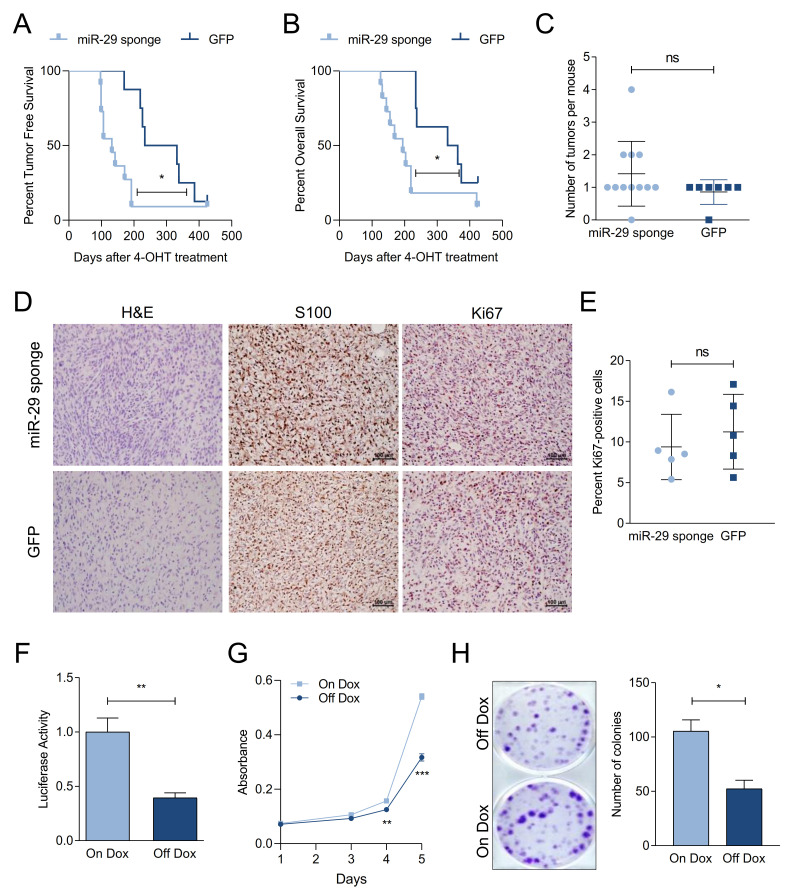
miR-29 inactivation promotes melanoma formation. (**A**,**B**) Kaplan–Meier curves comparing the tumor free survival (**A**) and overall survival (**B**) of Braf^V600E^; Pten^∆/WT^ GFP (*n* = 6) and miR-29 sponge (*n* = 10) chimeras using the Gehan-Breslow-Wilcoxon test; (**C**) number of melanomas that developed in the chimeras; (**D**) H&E (hematoxilin–eosin) staining and S100 and Ki67 immunohistochemistry on tumors from GFP and miR-29 sponge mice at endpoint. Bars indicate 100 µm; (**E**) quantification of Ki67-positive nuclei per field in tumors from GFP and miR-29 sponge mice; (**F**) miR-29 MRE-Luciferase reporter activity in miR-29 sponge melanoma cells. Dox withdrawal turns off expression of the sponge construct, resulting in endogenous miR-29 reactivation and reporter repression. The combined mean ± SEM of three independent experiments performed in quadruplicates is shown; (**G**,**H**) proliferation (**G**) and colony formation (**H**) upon miR-29 reactivation in miR-29 sponge melanoma cells. The mean ± SEM of one representative out of three independent experiments performed in quadruplicates is shown. ns, not significant; * *p* < 0.05; ** *p* < 0.01; *** *p* < 0.001.

**Figure 3 cancers-13-01408-f003:**
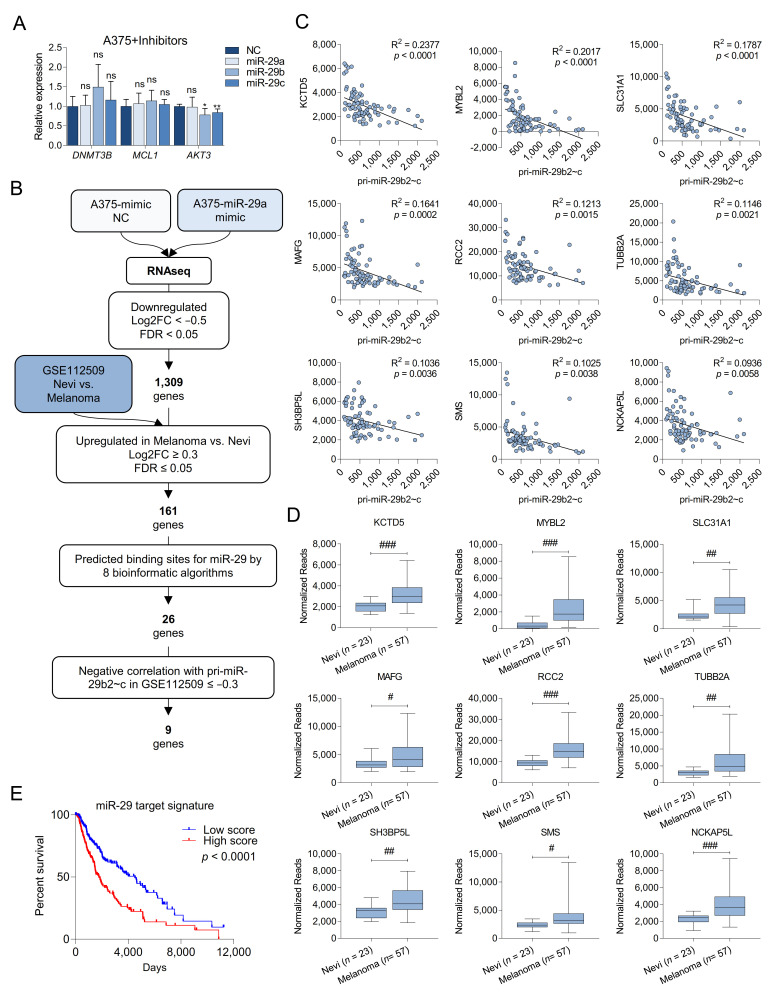
Identification of new miR-29 targets in melanoma. (**A**) Quantification by qRT-PCR of three validated miR-29 target genes of miR-29 after overexpression of miR-29 mimics in A375 cells. Cells overexpressing a negative control mimic were used as control; (**B**) selection flowchart to identify target genes of miR-29 involved in melanoma progression; (**C**) correlation between the normalized reads of pri-miR-29b2~c and the normalized reads of the top nine identified putative miR-29 targets in the GSE112509 dataset; (**D**) expression of the top nine identified putative miR-29 targets in 23 nevi and 57 melanomas obtained from the GSE112509 dataset; (**E**) survival analysis from TCGA (PanCancer Atlas, *n* = 363) comparing melanoma patients with high or low expression of the nine putative miR-29 targets. ns, not significant; * *p* < 0.05; ** *p* < 0.01; FDR, false discovery rate; # FDR < 0.05; ## FDR < 0.01; ### FDR < 0.001.

**Figure 4 cancers-13-01408-f004:**
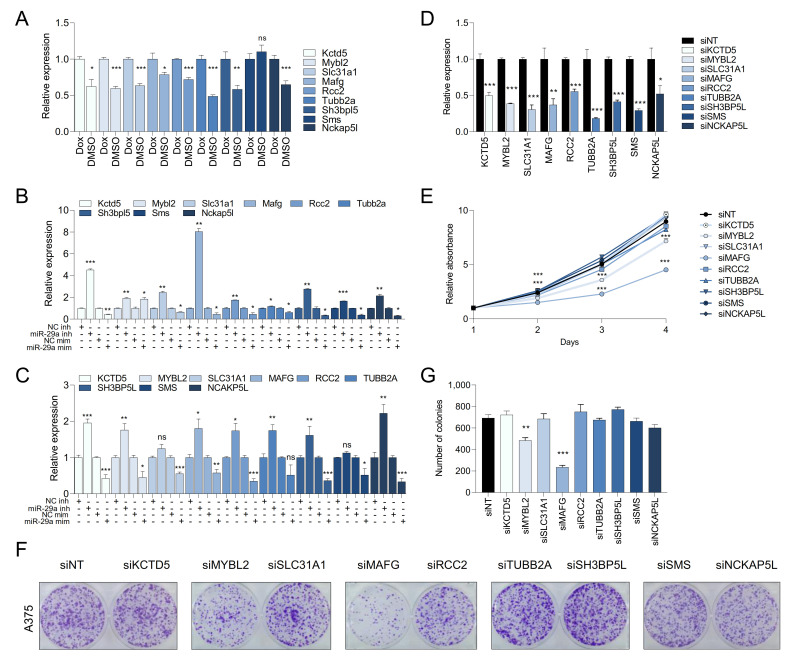
miR-29 targets constitute a vulnerability of melanoma. (**A**) expression of the nine putative miR-29 target genes upon miR-29 reactivation in miR-29 sponge melanoma cells; (**B**,**C**) expression of the nine putative miR-29 target genes upon miR-29 inhibitor or mimic transfection in miR-29 sponge melanoma cells (**B**) or human A375 melanoma cells (**C**). The mean ± SEM of one representative out of three independent experiments performed in triplicates is shown. Gene expression levels are normalized to β-Actin (mouse) or GAPDH (human); (**D**) validation of the efficacy of the ON-TARGETplus siRNA pool for the nine identified putative miR-29 targets; (**E**) effect of silencing the nine putative miR-29 targets on proliferation of A375 cells; (**F**) effect of silencing the nine putative miR-29 targets on focus formation of A375 cells; (**G**) quantification of colony formation shown in (**F**). The mean ± SEM of one representative out of two independent experiments performed in quadruplicates is shown. ns, not significant; * *p* < 0.05; ** *p* < 0.01; *** *p* < 0.001.

**Figure 5 cancers-13-01408-f005:**
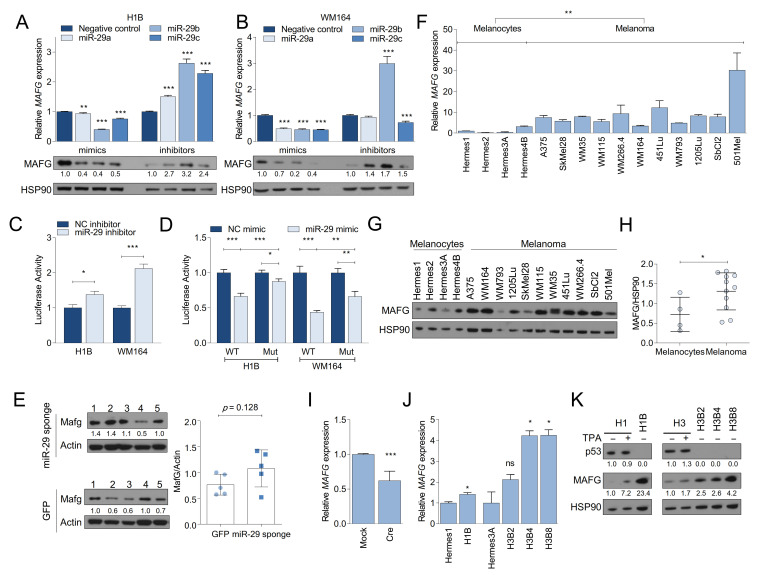
MAFG is a target of miR-29 and deregulated in melanoma. (**A**,**B**) MAFG mRNA (upper panels) and protein (lower panels) expression in response to transfection with miR-29 mimics (left) or inhibitors (right) in H1B melanocytes (**A**) or WM164 melanoma cells (**B**). The mean ± SEM of one representative out of two independent experiments performed in triplicates is shown. mRNA expression is normalized to GAPDH; (**C**) activity of MAFG 3′UTR Luciferase reporter in response to miR-29 inhibitors; (**D**) activity of MAFG wildtype or miR-29 binding site-mutant 3′UTR Luciferase reporter in response to miR-29 mimics. For (**C**,**D**), the combined mean ± SEM of two independent experiments performed in quadruplicates is shown; (**E**) Western blot showing MAFG expression in GFP and miR-29 sponge melanomas isolated from Braf^V600E^; Pten^Δ/WT^ chimeras; (**F**) qRT-PCR showing the basal expression levels of *MAFG* mRNA in melanocytes and melanoma cells. The mean ± SEM of one representative out of two independent experiments performed in triplicates is shown. Expression is normalized to GAPDH; (**G**) Western blot showing the expression levels of MAFG protein in melanocytes (*n* = 4) and melanoma cells (*n* = 11); (**H**) Quantification of the Western blot shown in (**G**); (**I**) qRT-PCRs showing the expression of *MAFG* in LSL-Braf^V600E^ PMM following Adeno-Cre/Mock infection; (**J**) qRT-PCR showing the expression levels of *MAFG* mRNA in parental and BRAF^V600E^-mutant melanocytes. The mean ± SEM of one representative out of two independent experiments performed in triplicates is shown. Expression is normalized to GAPDH; (**K**) Western blot showing MAFG expression in human melanocytes in response to TPA stimulation or chronic BRAF^V600E^ expression. The mean ± SEM of one representative out of two independent experiments performed in triplicates is shown. All Western blots show the intensity ratio of the protein of interest normalized to HSP90 or Actin. * *p* < 0.05; ** *p* < 0.01; *** *p* < 0.001.

## Data Availability

Datasets related to this article can be found as PRJNA624657 hosted at https://www.ncbi.nlm.nih.gov/sra.

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
