# Peer review of "A MAPK/miR-29 Axis Suppresses Melanoma by Targeting MAFG and MYBL2"

_cancers, 2021, doi:10.3390/cancers13061408_

Round 1
Reviewer 1 Report
An interesting experimental paper exploring the role of Mir 29 family of RNA in melanoma development. Only minor queries:
In the statistical analysis subsection, please specify the maker of the statistical information program and its location.
The conclusion paragraph should be expanded, with the future prospective and possible uses of this research's results.
Thank You
Author Response
Please, see the attachment

Reviewer 2 Report
The manuscript covers interesting research on the MAPK / miR-29 axis that in melanoma cell lines control the expression of transcription factors MAFG and MYBL2 responsible for melanoma progression. However, the manuscript suffers several flaws that need to be addressed.
The use of Hsp90 protein as a loading control for Western Blotting is rather unfortunate, especially that in several blots (e.g. in figures 1E, 5B(inhibitor part) and 5G) the uneven distribution of band intensity is clearly visible. Other blots are normalized to actin level, which is the most common protein for Western Blot normalization. What was the rationale of using Hsp90 in some cases? The authors should consider to unify loading control normalization, or at least provide better loading controls for forementioned blots.
The authors should add and summarize the information regarding the mutation state of melanoma cell lines used in the study (eg. BRAF, NRAS, p53 status) to easier draw conclusions from the results.
What are the units in figure 3C – the correlation plots? Is it also normalized reads as in figure 3D? Please specify that in the figure or its legend.
The first paragraph of the 2.4. subchapter should be moved to the end of the previous subchapter.
There is no figure 7H in the manuscript (line 368).
Unlike the authors state in the end of discussion section, MAFG transcription factor has been already linked to melanoma, and the paper (Fang et al, 2016) has been already cited as [24]. Please modify the discussion.
Author Response
Please, see the attachment.
